# Fabrication and Characterization of a Stretchable Sodium Alginate Hydrogel Patch Combined with Silicon Nitride and Metalized Halloysite Nanotubes to Develop a Chronic Wound Healing Treatment

**DOI:** 10.3390/ijms26041734

**Published:** 2025-02-18

**Authors:** Femi B. Alakija, David K. Mills

**Affiliations:** 1Molecular Science and Nanotechnology, Louisiana Tech University, Ruston, LA 71272, USA; 2School of Biological Sciences, Louisiana Tech University, Ruston, LA 71272, USA

**Keywords:** hydrogels, halloysite nanotube, nanocomposite, silicon nitride, antimicrobial properties

## Abstract

The human body is known as a responsive healing machine, but sometimes, broken bones do not heal, especially if a bacterial infection is present. The present study describes the fabrication and characterization of a nanocomposite hydrogel patch incorporated with silicon nitride and magnesium oxide (MgO) deposited on the halloysite nanotube (HNT) surface using a facile and inexpensive electrodeposition coating process. Scanning electron microscopy (SEM) was used to observe the surface morphology of the MgO/HNT surface coating and the nanocomposite patch. Material characterization, including SEM, contact angle, pore size analysis, and tensile properties, was performed to determine the composite’s structure and material properties. *E. coli* and *S. aureus* bacterial cultures were used to test the antimicrobial properties. Cellular response to MgO/HNTs was studied using mouse embryonic fibroblasts. The nanocomposite hydrogel patch was discovered to possess inherent properties when tested against bacterial cultures, and it was found to enhance fibroblast cell migration and proliferation. The nanocomposite hydrogel patch also showed sustained drug release. Materials involved in the fabrication helped in the swelling properties by which the nanocomposite hydrogel patch has approximately 400% of its initial weight discovered during the swelling test.

## 1. Introduction

Chronic nonhealing wounds place a significant burden on a patient’s quality of life and a sizeable financial strain on healthcare providers, especially as the treatment options are limited. Wounds that are not taken care of can become life-threatening. Depending on the injury’s severity, if pathogens are not cleared, the immune system’s strength is further invaded, which impedes wound healing. During the four stages of wound healing, it is essential to maintain an adequate environment (trace elements, minerals, vitamins) so the cells can achieve their function and heal well.

However, certain bacteria invade a wound using those nutrients and the moisture at the wound site to proliferate [1,2]. As a result, there is a restriction in the number of nutrients available for the healing cells, which makes bacteria invade a wound site, causing the wound to become chronic and more challenging to treat [3]. Antibiotics are good antimicrobial agents when used in wound dressing to fight bacterial infection, as they have many different targets depending on the antibiotic used [4,5]. Antibiotics can target the cell wall, the synthesis of nucleic acids, protein synthesis, and some metabolic pathways. However, most bacteria (70%) found in wound infections resist at least one antibiotic [6,7].

The primary goal of wound healing technology is to promote rapid and efficient healing [8,9]; it should enhance the body’s natural healing processes by providing a conducive environment for tissue regeneration [10]. This can be achieved by promoting new blood vessel growth and accelerating tissue production [11]. Infection is a significant risk factor in wound healing, particularly in cases where the wound is open to the environment. Therefore, wound healing technology should include measures to reduce the risk of infection [12]. This can be achieved through the use of antimicrobial agents or by designing the technology to prevent bacterial colonization [13,14].

Additionally, wound healing technologies should be minimally invasive to reduce the risk of complications and promote patient comfort [15,16]. Technologies that can be applied externally or through minimally invasive procedures, such as injections or topicals, would be ideal [15,16,17,18]. Also, wound healing technology should be safe and biocompatible. It should not cause any adverse reactions or toxicity, which is particularly important for long-term use, where the risk of adverse events may increase [19,20,21]. Due to their high water content, load-bearing biological tissues, and excellent biocompatibility, hydrogels have been used for wound healing, tissue/cartilage replacement, and scaffolds for cell growth [22]. When combined with electronics, hydrogels have been used to develop various biomedical devices, sensors, and switchable micropatterns [23].

Nanotechnologists have developed and implemented nanometer-sized components and how these can be directly delivered to a specific location [24]. In the case of some ailments, many drugs must be delivered to a targeted location where nanoparticles can be used to achieve this [25,26]. They are used as drug delivery systems where therapeutic agents/drugs are delivered directly to a specific location [24]. Some nanoparticles used in drug and gene delivery include ceramic nanoparticles made from silica and alumina [27].

Despite the benefit of the particles serving as a drug delivery system, their migration into treated organisms can pose significant risks. Some potential undesirable effects, if nanoparticles migrate into tissues, can accumulate over time, leading to possible long-term toxicity. This bioaccumulation can be particularly concerning for organs like the liver, kidneys, and brain [28,29]. Exposure to nanoparticles can also lead to neurotoxicity, damaging sensory neurons and receptors, which affects the central nervous system; this includes oxidative stress, immune suppression, and inflammatory response [30]. Studies have also shown the ecotoxicity of metal-based nanoparticles, encompassing exposure pathways and toxic effects on organs and tissue [31].

Certain metals and metal nanoparticles can provide antibacterial properties as an alternative to antibiotics. Unlike antibiotics, bacteria are less likely to develop resistance against metals [3]. Inorganic-based nanomaterials can either comprise metal or non-metal elements. They could also consist of an oxide form, hydroxide group, or phosphate group of the metal or nonmetal. These materials are generally used in biomedical devices, electronics, and sensors. Examples include silver (Ag), copper (Cu), copper oxide (CuO), aluminum (Al), magnesium oxide (MgO), gold (Au), zinc (Zn), zinc oxide (ZnO), iron (Fe), and iron oxide (Fe_2_O_3_, Fe_3_O_4_) [32].

Several studies have also demonstrated the effectiveness of metals and metal oxides such as silver (Ag), gold (Au), titanium oxide (TiO2), copper oxide (CuO), Zinc oxide (ZnO), and magnesium oxide (MgO) against gram-positive and gram-negative bacteria [33]. For instance, Silver (Ag) has outstanding physical, biological, and mechanical properties. As a result, Ag can eradicate infections during wound healing against *Escherichia coli*, *Pseudomonas aeruginosa*, and *Staphylococcus aureus*. Furthermore, the growth of pathogenic viruses and spoilage-causing fungi such as *Penicillium brevicompactum*, *Aspergillus versicolor*, *Candida tropicalis*, *Fusarium* sp. [34], as well as respiratory syncytial virus, herpes simplex virus [35], different strains of influenza virus, HIV-1, and monkeypox virus [33,34], *Pseudomonas aeruginosa*, *Staphylococcus aureus*, and *Escherichia coli* [36] have also been shown to be inhibited by the metal nanoparticle. 

Silicon nitride (Si_3_N_4_), a ceramic class similar to steel, is a synthetic, industrial, and non-oxide ceramic used for high-performance applications such as heat exchangers, gas turbines, and automotive engines due to its excellent temperature resistance, superior mechanical strength, relatively high toughness, and abrasion resistance [37]. In addition, bulk silicon nitride (Si_3_N_4_) is biocompatible, osteoconductive [38,39,40], has excellent corrosion resistance [41,42], and possesses antibacterial properties.

Si_3_N_4_ has been used to fabricate spinal fusion devices and intervertebral discs [41,43]. It has also been demonstrated that reinforcing Si_3_N_4_ ceramics with another material can improve its mechanical properties. For example, it was discovered that Si_3_N_4_ possesses high fracture toughness when added to MgO. Similarly, doping Si_3_N_4_ with either ytterbium or yttrium oxides or Titanium nitride (TiN) particles shows tough mechanical properties [37,44].

Si_3_N_4_ has performed better as an antimicrobial agent than implant materials such as titanium alloys, PEEK, and steel, which are currently used [44]. For example, in a study reported by [45], Si_3_N_4_, PEEK, and titanium exposed to bacteria colonies indicated percent microbe counts of 0%, 88%, and 21%, respectively, showing Si_3_N_4_‘s superiority over the other implant materials. Furthermore, Si_3_N_4_ can quickly bind the serum proteins necessary for subsequent cell adhesion and proliferation [42,46]. Additionally, under physiological conditions, some functional groups, such as -NH_2_ and -OH, can spontaneously develop on Si_3_N_4_ surfaces, giving Si_3_N_4_ a high degree of hydrophilicity and the potential for surface functionalization [47,48].

Halloysite nanotubes (HNTs) are a type of nanomaterial that is naturally available and abundant in clay particles [49,50]. HNTs are double-layered aluminosilicate nanoclay with a predominantly hollow tubular structure. It has a chemical formula of Al_2_Si_2_O_5_(OH)_4_nH_2_O. HNTs have an external diameter of 50–80 nm, an inner lumen of 10–15 nm, and a length of 1 µm [51]. Each cylinder has concentric alternating layers of aluminum and silicate, giving the cylinder or tube buoyant and negatively charged layers [52,53].

Recently, there have been more applications of hydrogels in drug delivery, electronics, and nanomedicine due to the inclusion of nanoparticles in scaffolds because of their improved characteristics [54,55]. Hasnain and his colleague demonstrated other application areas of hydrogel, such as in food packaging, magnetic devices, sensors, wound dressing, and prosthetic devices due to metal nanoparticles, which provide mechanical support [56,57,58]. A biodegradable patch for cardiac tissue engineering was reinforced with titanium oxide (TiO_2_), as shown by [59], to have greater tensile strength than the one without TiO_2_.

Mouse Embryonic Fibroblasts (MEF) and bioactive nanoparticles provide an ideal platform for clinical tissue studies involving wound care treatment [60]. Applying hydrogels with encapsulated MEF to cover burn wounds would provide many immediate advantages, such as a protective layer from the outside environment. In addition, it would stimulate the proliferation and secretion profiles of vital growth factors for accelerating skin repair.

The research in this manuscript aims to fabricate a biodegradable nanocomposite hydrogel patch to facilitate chronic wound healing by incorporating silicon nitride and metalized halloysite nanotube. With the benefits of HNT and Si_3_N_4_ discussed above, we aim to electrodeposit MgO onto the outer surface of HNT to increase its properties. This will be used to fabricate and characterize a hydrogel patch, test with gram-positive and negative bacteria to know the effect without drugs, and study its impact in vitro on wound healing.

## 2. Results

### 2.1. Material Characterization

#### 2.1.1. MgHNTs and Nanocomposite Hydrogel Patch Surface Analysis

Digital microscope images of dry and wet samples are shown in Figure 1A,B, where Na-alginate is lighter than the rest, and a bigger space leading to a larger pore than the rest is seen in the dry image. SEM images shown in Figure 2 showed a thicker and better distribution of the nanocomposites, as shown in Figure 3. Na-alginate without any nanocomposite shows a lump-like irregular shape with a denser surface due to its polysaccharide chains’ clustering. However, the addition of Si_3_N_4_ shows clustering all over the Na-alginate surface. The addition of MgHNT to Na-alginate gives the structure a smoother and less porous shape.

#### 2.1.2. Surface Area and Pore Size Analysis

Brunauer- Emmett-Teller (BET) and Barrett-Jayner Halenda (BJH) analysis (for pore size distribution), as discussed in Section 4.4.2 uses Helium adsorption-desorption to distribute pore volume against pore size. Figure 4 shows a decrease in the nanocomposite pore size as more additives are added. The highest pore radius was Na-alginate at approximately 14.084 Å because it contains no nanocomposite, which is also evident in the swelling test shown in Figure 5. The resulting distribution of the pore sizes within the material will affect the material’s ability to retain or pass certain materials. Na-alginate has the largest pore size, rendering it unable to retain some materials, making its use in drug delivery difficult. However, other samples have smaller pore sizes and better distribution, as shown in the SEM image, which allows for sustained drug delivery.

#### 2.1.3. Swelling Behavior Analysis

The capability of hydrogel to absorb fluid is essential in wound dressing. The stability of the nanocomposite hydrogel patch was investigated by observing the weight changes at a time (t) after being subjected to PBS, as shown in Figure 5. All nanocomposite hydrogels showed an increase of more than 250% of their initial weight. Si_3_N_4_MgHNT-Na-alginate has the highest increase in weight among the nanocomposite hydrogels of more than 400% of its initial weight, while Si_3_N_4_-Na-alginate has up to 350%. Na-alginate without nanocomposite showed the highest of all samples of up to 460%, possibly due to spaces not occupied by nanocomposite. The high increase in weight was due to the presence of Na-alginate’s hydrophilic nature since it is a linear polysaccharide from alginic acid.

#### 2.1.4. Wettability Analysis

The contact angle of the nanocomposite hydrogels decreased as hydrophilicity increased after SBF treatment. However, as SBF was dropped on the Na-alginate/Si_3_N_4_MgHNT hydrogel’s surface, it spread out quickly compared to other nanocomposite hydrogels tested because its surface turned out to be hydrophilic due to the combination of Si_3_N_4_ and MgHNT. The Contact angle image is shown in Figure 6, while its graphical representation is shown in Figure 7. Since the contact angle provides insight into the wetting behavior of the hydrogel surface, a smaller contact angle of nanocomposite with Si_3_N_4_ indicates high wettability and hydrophobicity, which are desirable in drug delivery and wound dressings. This offers insight into the hydrogel’s performance in a wet environment.

#### 2.1.5. Tensile Properties

It was discovered in Figure 8 that the hydrogel without any reinforcement (Na-alginate) has weaker elongation (~23 mm) compared to the nanocomposite hydrogels. Na-alginate/MgHNT has approximately 34 mm before breakage. At the same time, nanocomposite hydrogels with Si_3_N_4_ do not break for 90 s of the experiment, having longer and better elongation than hydrogels without Si_3_N_4_. Na-alginate/Si_3_N_4_ has the better tensile strength of all hydrogels due to the reinforcement of only Si_3_N_4,_ providing greater strength. The red arrow points to the breaking point at which the material fractures due to it being stretched beyond its ultimate tensile strength, a point where it can no longer increase in applied stress or strain, cutting short its elongation. Adequate tensile strength ensures the hydrogel can maintain its structure under mechanical stress. The green arrow points to the elasticity behavior of the hydrogels.

### 2.2. Hydrogel Cellular Response

#### 2.2.1. Cell Viability Analysis

Control cells are cells without the samples, which act as a negative control. Control cells at the beginning of day 1 have a viability percentage of approximately 100% and decrease to 95% at the end of day 7. Na-alginate without any nanoparticles has a viability percentage of 100% on day 1 and reduces to 99% on day 7. Na-alginate/Si_3_N_4_ at the beginning of day 1 has a viability percentage of approximately 100% and 97.7% at the end of day 7. Na-alginate/MgHNT increased from 99% on day 1 to 99.2% on day 7. Similarly, Na-alginate/Si_3_N_4_MgHNT decreased in percent viability from 100% on day 1 to 98% on day 7. The images of the live and dead cells are shown in Figure 9. Graphs showing quantitative values of calculated cell count are shown in Figure 10.

#### 2.2.2. Cell Proliferation Assay

It is essential to confirm that the nanocomposite hydrogels will proliferate mouse embryonic fibroblast cells. Therefore, a proliferation assay on the hydrogels was performed for 7 days, as shown in Figure 11. Hydrogel without nanoparticles showed the lowest proliferation rate compared to other nanocomposite hydrogels and the control cells without additives. Although control cells have the highest proliferation before day 7, they have the lowest proliferation by day 7. The result indicated that sodium alginate promotes cell proliferation. Still, when combined with other biomaterials, it can enhance adhesion and proliferation, meaning that the introduction of nanoparticles to NA-alginate does not compromise its proliferation property. This shows that the nanocomposite hydrogel can support long-term cell viability and activity for clinical and industrial use.

#### 2.2.3. Wound Healing Analysis

The cellular behavior was observed by taking images with a 4-in-1 cell imaging revolve microscope (Echo-A bico company, San Diego, CA, USA). Figure 12 shows the way mouse embryonic fibroblast cells behaved, while the quantification is shown in Figure 13. Control cells and all nanocomposite hydrogels showed that 100% of cells in the wounded part migrated and closed the wound gap after 18 h. However, Na-alginate without nanocomposite and Na-alginate/MgHNT showed wound closure after 12 h. This signifies the role that Si_3_N_4_ played in how cells respond to injury, and thus, it can be concluded from this result that clustering affects the rate at which cells close wounds. However, all nanocomposite samples completed the cycle of closing the wound gap after an additional 6 h.

#### 2.2.4. MEF Migration Analysis

The migratory analysis is characterized in Figure 14. Negative control cells (mouse embryonic fibroblast cells) show a decrease in cell migration, which was expected due to the absence of nanocomposites acting as a chemoattractant. Na-alginate/Si_3_N_4_ and Na-alginate/Si_3_N_4_MgHNT show the highest overall rise in cell migration through the chamber pore. All nanocomposite hydrogels showed an increase in cell migration more than the positive control cells. This result indicates that with the addition of Si_3_N_4_ and Mg, there is an increase in the migration of the cells. However, Na-alginate without additional components supports cell migration slightly less than those with additional components. Still, the key point here is that additional components to Na-alginate provide other benefits, including the benefit of Na-alginate.

#### 2.2.5. Senescence Analysis

Images of SA-β-galactosidase-stained mouse embryonic fibroblast cells are shown in Figure 15, and the percentage of the SA-β-galactosidase-stained positive cells is shown in Figure 16. Na-alginate/Si_3_N_4_ and Na-alginate/Si_3_N_4_MgHNT nanocomposite fibers showed a low percentage of β-galactosidase positive cells (0.0083%) compared to other nanocomposite hydrogels and the control cells. Additionally, they showed uniform staining, suggesting that the hydrogel permits controlled diffusion, which is essential for drug delivery.

#### 2.2.6. Sustain Release of Gentamicin Sulfate Analysis

Samples with HNT were loaded with a drug (gentamicin) and sustained release over 96 h. Control is just PBS without drugs since drugs can only be loaded into the inner hollow of the HNT. The percentage of drug release from gentamicin-loaded nanocomposite hydrogels is shown in Figure 17. There was a burst release at the beginning of the experiment, from 0 h to around 40 h, followed by a more constant, stable release rate. However, most of the nanocomposite hydrogels followed mono-phasic kinetics, where the controlled release was more stable and constant. It can be seen that both samples containing HNT deliver drugs at a similar rate because they both have the same HNT sample before the addition of the other components. However, the release rate of the nanocomposite with Si_3_N_4_ is slightly higher than that without Si_3_N_4_.

### 2.3. Antimicrobial Susceptibility Analysis

At the end of 48 h, the bacteria culture was tested using the plate reader at 630 nm. The results of the bacterial inhibition are shown in Figure 18 and Figure 19, where only bacteria + media shows no inhibition throughout the 48 h. At the same time, all other samples, including the control (Gentamicin + bacteria), show reduced inhibition. In addition, it was discovered that in 0.5 mg of Na-alginate/MgHNT, there was increased *S. aureus* growth for 24 h before a constant growth for an additional 12 h, and then increased growth showed no bacteria inhibition.

## 3. Discussion

Wound dressing or patch must be able to eliminate continuous bleeding and any obstruction of cell proliferation. It should be able to regulate tissue remodeling and avoid inflammation [61]. Diabetic wounds, for example, have been reported to have infection risk and tissue reconstruction problems [62]. Therefore, a wound dressing or patch is needed to help protect the wound from infection, accelerate wound healing, and increase angiogenesis. Under physiological conditions, alginate is seen to have a slow degradation rate and is nonbioactive [63]. Therefore, sodium alginate was added to silicon nitride and metalized halloysite nanotubes, as seen in Figure 20, to increase its degradation rate, antimicrobial properties, tensile properties, and wound closure.

An adequate wound patch should have swelling properties capable of holding moisture from wounds. It was discovered that the fabricated hydrogel patch has approximately 400% of its initial weight. Although hydrogel without any nanocomposite has better-swelling properties due to the large space, as shown in Figure 5, this also corresponds to the highest pore size of all hydrogels. The relationship between the hydrogel composition and the contact angle was investigated. It has been documented that an increase in the degree of diffusion leads to a decrease in contact angle [64], which is evident in Figure 6 as SBF spreads out very fast as it is dropped onto the Na-alginate/Si3N4MgHNT surface, decreasing the hydrophilicity. Similarly, nanocomposite hydrogels have better tensile strength and elongation than hydrogels without reinforcement.

Mouse embryonic fibroblasts (MEFs) are one of the cells used to study wound healing and have structures like stem cells [65]. At the end of day 7, the proliferation assay shown in Figure 11 shows that all nanocomposite hydrogels have a better proliferation rate than the control cells. The cytotoxicity test indicated that the hydrogels are non-toxic. In contrast, all hydrogels increased wound closure after 18 h during the scratch assay, indicating a self-healing property, while the bacteria test shows its bacterial properties. The ability of the wound healing patch to deliver drugs was investigated during the drug release study, which showed a more stable, constant release rate of gentamicin into PBS.

Silicon Nitride applications in wound healing were explored due to its biocompatibility, antibacterial properties, and mechanical strength. It is an excellent candidate for medical patches that promote healing and prevent infections. The ability of Silicon nitride to support cell growth while resisting bacterial growth is especially beneficial in treating chronic or complex wounds. On the other hand, Halloysite nanotubes offer unique characteristics due to their inner layer being capable of delivering drugs and other therapeutic agents. When metalized by adding magnesium, these nanotubes exhibit increased antimicrobial properties, further helping prevent infections in wound sites. Additionally, MgHNT nanoscale dimensions and compatibility with biological tissues make them ideal for advanced wound dressings and drug delivery systems. The fabricated hydrogel patch provides additional benefits that allow cell migration and the ability to absorb wound exudate through the swelling test, protecting the wound from external contaminants. Together, these materials represent innovative solutions in the field of regenerative medicine, opening doors to more effective and efficient wound management strategies.

## 4. Materials and Methods

### 4.1. Chemicals and Reagents

Magnesium oxide (CAS: 1309-48-4), Halloysite nanoclay (LOT: 685445-500G, CAS: 1332-58-7), Senescence cell histochemical staining kit (CAT: #CS0030-1KT), Gentamicin sulfate salt (LOT #049M4874V), Sodium alginate (LOT #SHBL 1627, CAS: 9005-38-3) were obtained from Sigma-Aldrich, St. Louis, MO, USA. Dulbecco’s Modified Eagle’s Medium (DMEM) (LOT: 2393824) and Penicillin/streptomycin (REF: 15070-063) were obtained from Gibco Invitrogen, Grand Island, NY, USA. Fetal bovine serum (FBS) (CAT: FBS002, LOT: N21H21) was obtained from Neuromics, Edima, MN. Cell counting kit-8 (CAT: #DJDB4000X) was from Vita scientific, College Park, MD, USA. The biotium viability/cytotoxicity assay kit (CAT: 30002-T, LOT: 210811) was from Biotium, Fremont, CA, USA. Silicon nitride (CAS; 12033-89-5) was obtained from 3DTech, Grand Rapids, MI, USA, and Chem savers, Bluefield, VA. Phosphate buffer saline (PBS; REF: 25-508B; LOT: MS00LB) was from Genesee scientific, San Diego, CA, USA. Mouse Embryonic fibroblast cells (MEF, SCRC-1040), *Escherichia coli* (ATCC 25922), and *Staphylococcus aureus* (ATCC 6538) were obtained from ATCC, Manassas, VA, USA. Cytoselect^TM^ 24-well cell migration assay, 8 µm, Fluorometric format (CAT: #CBA-101) was obtained from Cell Biolabs, San Diego, CA, USA. Acrylamide monomer (CAT NO. 164855000; LOT: A0421569), N,N^1^- methylenebisacrylamide (CAT NO. J66710, LOT: P11H040), Ammonium peroxydisulfate (CAT NO. 054106.22; LOT: Q191061) were all obtained from Thermoscientific, Waltham, MA, USA. The equipment used in this study includes a Scanning Electron Microscope (Hitachi S4800 Field Emission SEM, Tokyo, Japan) for imaging and surface analysis. A NOVA2200e Surface Area and Pore Size Analyzer (Quantachrome—Anton Paar, Boynton Beach, FL, USA) was utilized for porosity measurements. The Contact Angle System OCA 15plus (Future Digital Scientific Corp, Westbury, NY, USA) was employed to assess surface wettability. Mechanical testing was conducted using a Biomaterial Testing Unit (CellScale, Waterloo, ON, Canada). Additionally, cell counting was performed with the Auto T4 Cellometer (Nexcelom Bioscience, Lawrence, MA, USA). Scratch assay images were taken with a 4-in-1 cell imaging revolve microscope (Echo-A bico company, San Diego, CA, USA).

### 4.2. Preparation of Metalized HNT (MgHNT)

The electrodeposition of metalized Halloysite nanotube was done according to a previous protocol [66].

### 4.3. Fabrication of Biodegradable Nanocomposite Hydrogel Patch

Na-alginate (2 g) was dissolved in 100 mL water and stirred with a mechanical stirrer under 60 °C heating, as shown in Figure 20. For the samples with additional polymers, 1.0 g/10 mL Si_3_N_4_, MgHNT, or Si_3_N_4_MgHNT was added for extra toughness. Next, 15 g of Acrylamide (AAm) monomer was added to increase the thickness and gently moved around for even mixing. 0.1 g/10 mL H_2_O solution of N, N^1^-methylenebisacrylamide (MBAAm) crosslinker was added, while 1 g/10 mL H_2_O solution of Ammonium peroxydisulfate (initiator APS) added and 0.2 mL accelerator N, N, N, N-tetramethyl ethylenediamine (TEMED) was added to initiate polymerization reaction and left to allow to mix. When required, after 24 h, hydrogels were lyophilized.

### 4.4. Materials Characterization

#### 4.4.1. Morphology and Surface Characterization of MgHNTs and Nanocomposite Hydrogel Patch

A Scanning Electron Microscope (Hitachi S4800 Field Emission SEM, Tokyo, Japan) was used to study the surface morphologies of the nanocomposite patch. The hydrogel samples were freeze-dried to avoid issues like water evaporation under a vacuum. Also, since hydrogels are non-conductive, they are coated with a thin layer of Gold to prevent charging under the electron beam before viewing under SEM.

#### 4.4.2. Multi BET/Pore Size Testing

NOVA2200e Surface area and pore size analyzer (Quantachrome-Anton Paar, Boynton Beach, FL, USA) was used to determine the pore size of the nanocomposite hydrogels. The lyophilized hydrogel was dried and weighed (average weight of approximately 1.1391 g) before being analyzed. The outgas temperature was 70 °C for 24 h, and the helium adsorption-desorption method was used.

#### 4.4.3. Swelling Test

2000 mg of nanocomposite hydrogels were weighed as the initial weight, placed in a 15 mL tube containing 10 mL PBS, and then incubated at 25 °C for 24 h. After every 4 h, the samples were taken out of PBS, excess PBS was removed with filter paper, and then the final weight measurements were taken. The % swelling was calculated using Equation (1). All sample swelling tests were done in triplicate.(1)$$\frac{Wft-Wi}{Wi}\times 100=Swelling\;(\%)$$
where: *Wft*: Final weight of the incubated sample at time *t*, *Wi*: Initial weight of the samples

#### 4.4.4. Contact Angle Measurement

The hydrophilicity of the nanocomposite hydrogels was determined using the contact angle system OCA (Future Digital Scientific Corp, model: OCA 15plus, Westbury, NY, USA). This was performed using the sessile drop method in triplicates using Simulated body fluid (SBF).

#### 4.4.5. Tensile Properties Testing

Na-alginate hydrogel and the nanocomposite hydrogels were poured into the same type of plate to polymerize for 24 h and cut into equal sizes after polymerization. Hydrogel’s elongation (ε) and tensile strength (σ) were determined with the Biomaterial Testing Unit (CellScale, Waterloo, ON, Canada) with 200 N load cells with a 1 mm/min speed and set up according to Figure 21.

### 4.5. Evaluation of In Vitro Fibroblast Response

#### 4.5.1. Cell Culture and Culture Medium

DMEM was used, then 10% fetal bovine serum (FBS-Neuromics, Cat No. FBS002, Lot No. N21H21, Edina, MN, USA) and 1% penicillin/streptomycin antibiotic (Gibco, REF. 15070-063, Billings, MT, USA) was added to make complete media and filtered for contamination. Cryopreserved mouse embryonic fibroblast (MEF) was obtained from ATCC.

#### 4.5.2. Conditioning of Nanocomposite Hydrogels

Dried nanocomposite hydrogels were weighed at approximately 163–165 mg, sterilized with UV light, and added to 5 mL of DMEM, incubated at 37 °C. 5 mL of DMEM was added to the extracted samples after 72 h and mixed by a vortex.

#### 4.5.3. Proliferation Assay

90–95% confluency MEF cells were detached, counted with Nexcelon Bioscience Auto T4 Cellometer cell counter, and cultured with a conditioned sample in a 12-well culture plate at 37 °C and 5% CO_2_ level in a humidified incubator. The cell counting kit-8 reagent (Vita scientific Cat #DJDB4000X, Sydney, Australia) was thawed and added to the cell culture plate (1/10 µL/well of total cells) at 1, 3, 5, and 7 days, incubated for 1–4 h at 37 °C. A Microplate reader was used to record the optical density at 450 nm.

#### 4.5.4. Cytotoxicity Assay

Conditioned samples were evaluated for cell viability using a biotium viability assay kit (Cat. 30002-T, Lot. 210811) for 1, 3, 5, and 7 days. The reagent staining solution contained 5 µL, 20 µL, and 10 mL of 4 mM calcein, 2 mM EthD, and PBS, respectively. 500 µL of cells/well were cultured with a conditioned sample in a 24-well plate and incubated at 37 °C and 5% CO_2_ in a humidified incubator. Cells were allowed to reach 90–95% confluency, then washed with 300 µL PBS twice before staining with the reagent staining solution and incubated at room temperature for 30–45 min. Fluorescence images of the cells were taken and counted with Image J (https://imagej.net/ij/).

#### 4.5.5. Scratch Assay

MEF cells were counted and cultured in a 24-well plate and were allowed to attain confluency before introducing a scratch with a 200 µL pipette tip. Detached cells were washed away during scratching before culturing the culture plate with a conditioned sample. Cells were analyzed for 24 h at 6 h intervals while % wound area was calculated according to Equation (2).(2)$$\frac{Mi}{Mt}\times 100=wound\;closure\;(\%)$$
where: *Mi*: Area of wound after scratching, *Mt*: Area of the wound at time *t*.

#### 4.5.6. Migration Assay

The ability of MEF cells to migrate toward a chemoattractant was evaluated. Cells were counted (1.0 × 10^6^), and 300 µL cells/mL were seeded in a serum-free media in the insert while 500 µL of chemoattractant (nanocomposite hydrogels) were added to the lower chamber of the migration plate (Cytoselect^TM^ 24-well cell migration assay, Cell Biolabs Cat. #CBA-101, San Diego, CA, USA). The plate was incubated at 37 °C and 5% CO_2_ level for 24 h. 225 µL cell detachment solution was added to a well containing the insert and incubated for 30 min at 37 °C. The cells that migrated towards the underside of the membrane were dislodged in the cell detachment solution before being incubated in the staining solution (4× Lysis buffer/CyQuant^®^GR dye solution) for 20 min. A microplate reader was used to record the absorbance at 490 nm.

#### 4.5.7. β- Galactosidase Staining

Mouse embryonic fibroblast cells expressing β-galactosidase were evaluated according to the manufacturer’s protocol (Sigma-Aldrich Cat. #CS0030-1KT, St. Louis, MO, USA) in the presence of conditioned nanocomposite hydrogels. MEF cells were seeded, cultured, incubated, and allowed to reach confluency and then fixed in 1X fixation buffer for 6–7 min at room temperature, stained, and then incubated at CO2-free 37 °C overnight.

### 4.6. In Vitro Drug Release Study

The vacuum entrapment method was used to load gentamicin sulfate into HNT before electrocoating with Magnesium to create MgHNT. 170–173 mg of gentamicin-loaded nanocomposite hydrogels were placed in 10 mL PBS and incubated at 37 °C with shaking. 1 mL of the release solution was taken every 12 h from the solution tube and replaced with the same volume of PBS. A microplate reader was used to evaluate the release solution for 96 h at 405 nm. The concentration was calculated using a standard calibration curve.

### 4.7. Antimicrobial Testing of Nanocomposite Hydrogel Patch

Antimicrobial testing was conducted by culturing lyophilized hydrogels and nanocomposite hydrogels with *E. coli* and *S. aureus* for 48 h at 37 °C in different concentrations of 2 mg, 1 mg, and 0.5 mg with tryptic soy broth and Mueller Hinton agar, while bacteria and media without the samples serve as the negative control and bacteria with gentamicin, serves as the positive control. A Microplate reader was used to take the optical density (turbidity of the broth) at 0, 12, 24, 36, and 48 h at 630 nm.

### 4.8. Statistical Analysis

Unless otherwise indicated, all results were reported as mean ± standard deviation (*p* < 0.05, *n* = 3). Statistical analysis was performed using Microsoft Excel (Office 16) with one-way analysis of variance (ANOVA) to determine differences between samples. Therefore, a level of *p* < 0.05 was considered statistically significant.

## 5. Conclusions

The study demonstrates the successful fabrication of wound healing patches using Na-alginate, Si_3_N_4_, and MgNHT, presenting a promising approach to enhanced wound care. The ionic nature of Na-alginate influences its interaction with cells, and when crosslinked, it provides mechanical support. Therefore, incorporating SNMgHNT into the Na-alginate matrix improved its mechanical strength, swelling capacity, and wettability while providing a sustained release of therapeutic agents. These properties are critical for monitoring a moist wound environment, promoting cellular proliferation, and preventing infections.

Our results indicate how the synergistic interaction between Na-alginate and SNMgHNT can optimize the physical and biological characteristics required for effective wound healing. The fabricated patches exhibited notably antimicrobial properties against gram-negative and positive bacteria, effective drug loading capacity, and controlled drug release, making them suitable for chronic wound treatment. Additionally, future studies will focus on clinical evaluations to further validate these patches’ therapeutic efficacy and safety in vivo.

Na-alginate without the addition of SNMgHNT shows some good properties. Still, a significant improvement is seen from our fabricated hydrogel patch, which is capable of expressing multifunctional properties such as drug delivery, antimicrobial properties, and mechanical properties.

Some of the limitations of this study include scaling up the fabrication process while maintaining consistency in the properties of the wound patches, as well as sterilizing the wound healing patches without damaging the active components or altering the properties. Additionally, obtaining approval from a regulatory body for the novel device may involve extensive testing to ensure safety, efficacy, and compliance with medical standards.

## Figures and Tables

**Figure 1 ijms-26-01734-f001:**
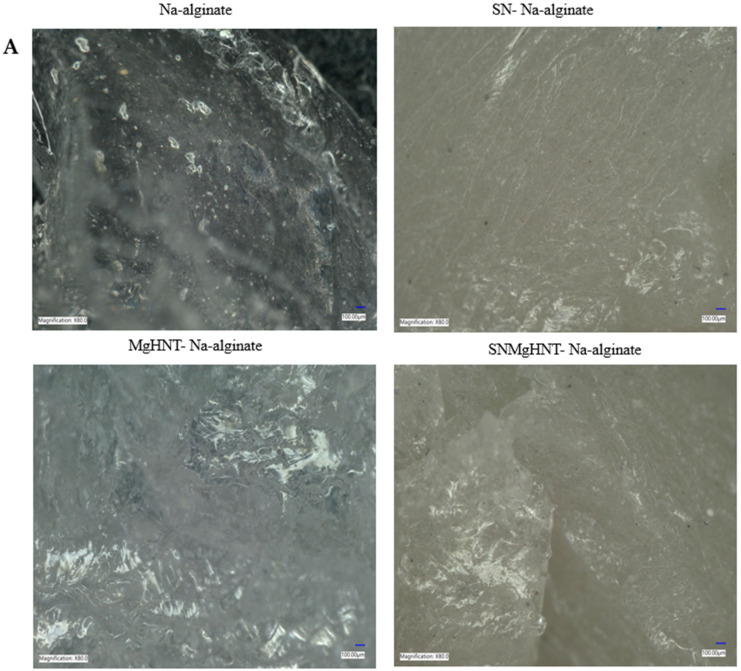
Digital Microscope Images of Surface Morphology of (**A**) Wet Nanocomposite Hydrogels with magnification of ×80.0 (**B**) Dry Nanocomposite Hydrogels with magnification of ×150.0.

**Figure 2 ijms-26-01734-f002:**
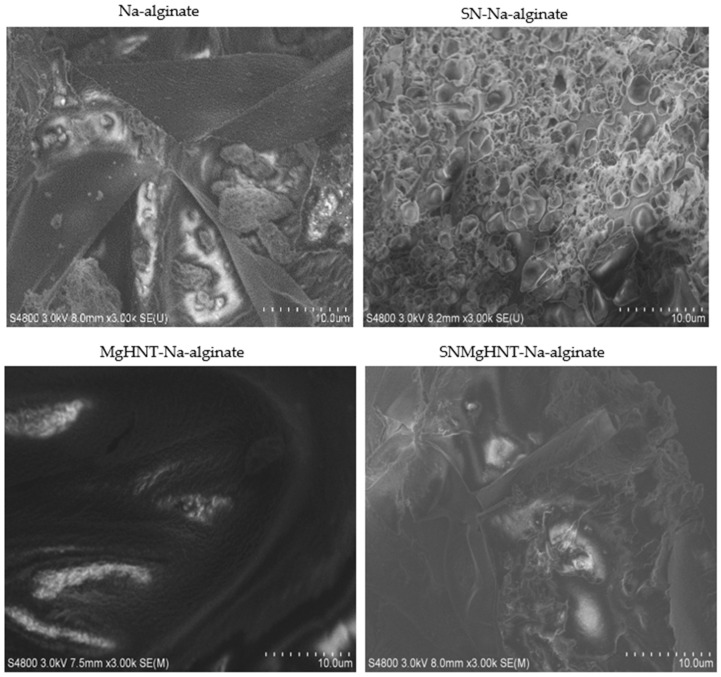
SEM Images of Surface Morphology of Nanocomposite Hydrogels.

**Figure 3 ijms-26-01734-f003:**
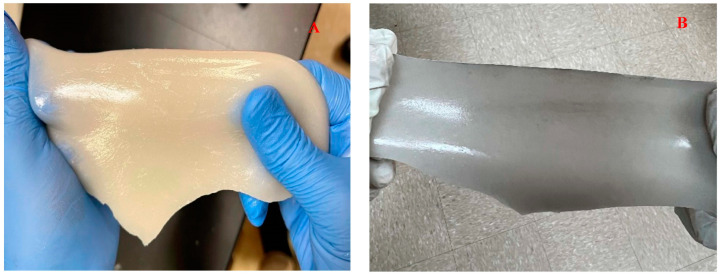
Picture of (**A**) Si_3_N_4_MgHNT and (**B**) Si_3_N_4_ Hydrogel Patch.

**Figure 4 ijms-26-01734-f004:**
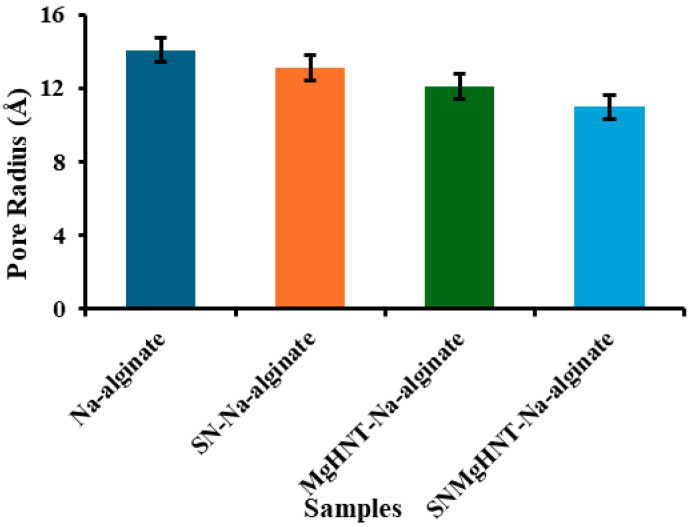
Average Pore Radius (Å) of all nanocomposite hydrogels, Measured by NOVA 2200e Surface Area and Pore Analyzer. Error Bars are Standard Deviations, where *n* = 3.

**Figure 5 ijms-26-01734-f005:**
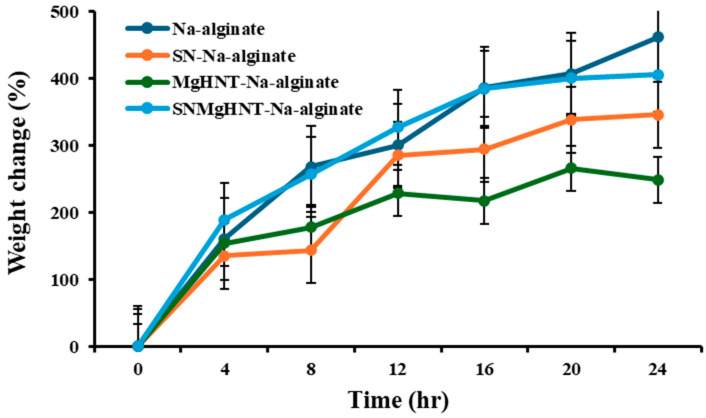
Weight change (%) of all Nanocomposite Hydrogels. Error Bars are Standard Deviations where *n* = 3.

**Figure 6 ijms-26-01734-f006:**
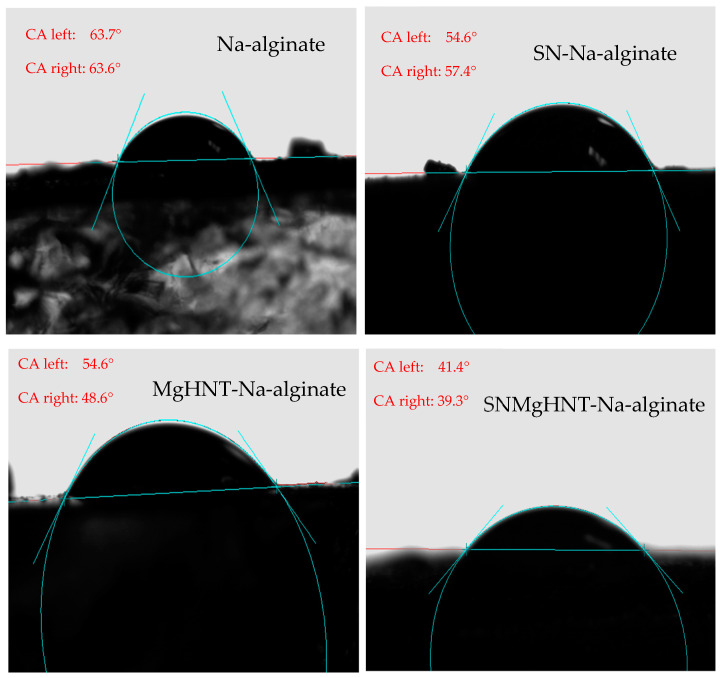
Image of Nanocomposite Hydrogels’ Contact Angle. CA left and CA right refer to the contact angles measured at the left and right sides of the SBF droplet placed on the hydrogel. These values help in evaluating the symmetry of the droplet and the uniformity of the surface. For instance, a symmetric droplet (CA left ≈ CA right) suggests a homogenous and level surface while an asymmetric droplet (CA left ≠ CA right) might indicate surface roughness, or inclination of the hydrogel.

**Figure 7 ijms-26-01734-f007:**
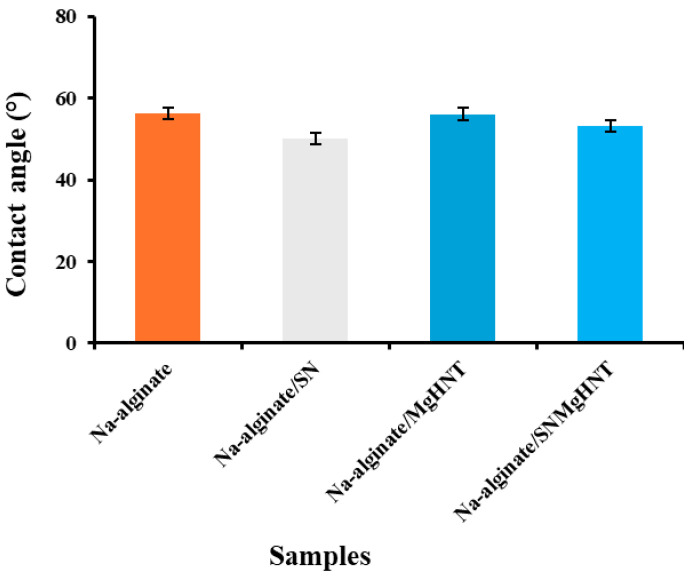
Nanocomposite Hydrogels Contact Angle Measurement. Error Bars are Standard Deviations where *n* = 3.

**Figure 8 ijms-26-01734-f008:**
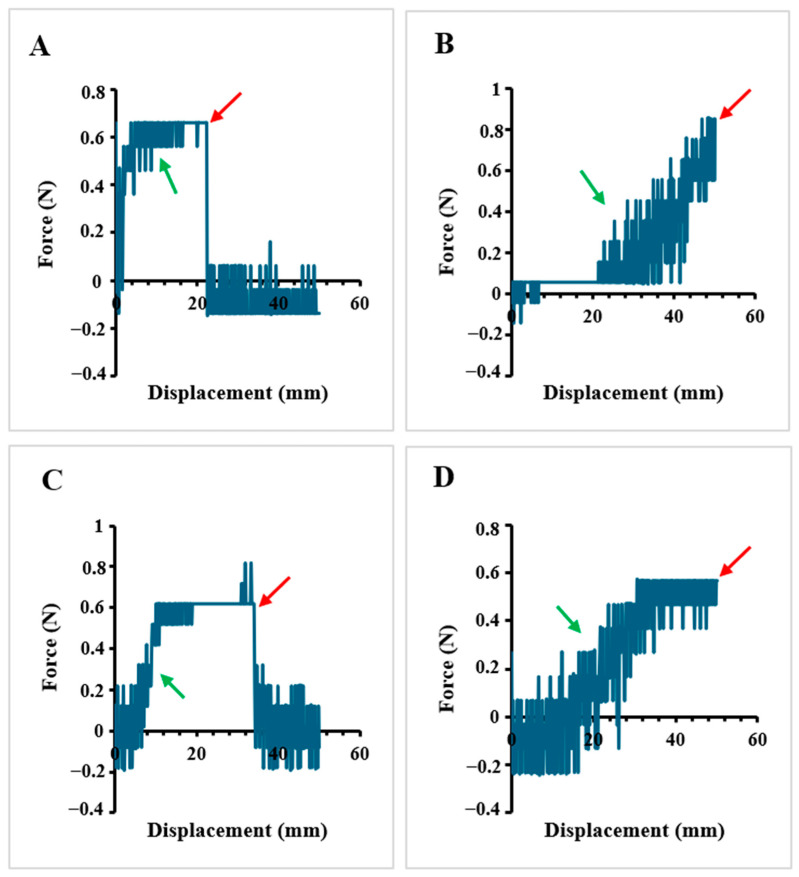
Tensile Test of (**A**) Na-alginate Hydrogel, (**B**) Na-alginate/Si_3_N_4_ Hydrogel, (**C**) Na-alginate/MgHNT Hydrogel (**D**) Na-alginate/Si_3_N_4_MgHNT Hydrogel. The red arrow points to the breaking point at which the material fractures, while the green arrow points to the elasticity behavior of the hydrogel.

**Figure 9 ijms-26-01734-f009:**
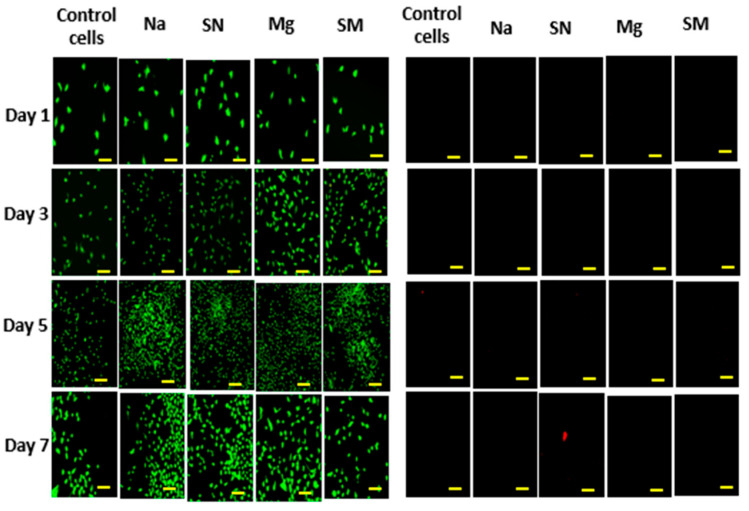
Images of Cytotoxicity Test (Livedead Assay) with Mouse Embryonic Fibroblast Cell of Nanocomposite Hydrogels. Live Cells in Green (**Left** Image) and Dead Cells in Red or Black Indicate no Dead Cell (**Right** Image) for day 1, 3, 5, and 7; Na = sodium alginate, SN = Na-alginate/Si_3_N_4_, Mg = Na-alginate/MgHNT, SM = Na-alginate/Si_3_N_4_MgHNT. Scale bar is 350 µm.

**Figure 10 ijms-26-01734-f010:**
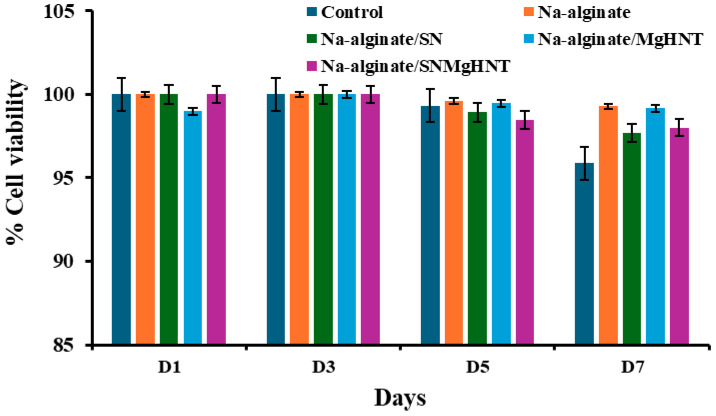
Graph of Nanocomposite Hydrogels Showing Quantitative Cell Count Value Calculated (Live Cells/Total Cell Count). Error Bars are Standard Deviations, where *n* = 3.

**Figure 11 ijms-26-01734-f011:**
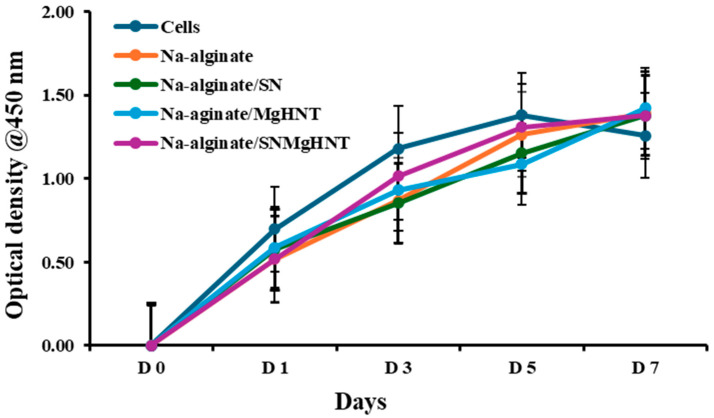
Proliferation Assay for Nanocomposite Hydrogels after Exposure to Mouse Embryonic Fibroblast Cells for 7 days. The Blue Line Signifies Control Cells. Error Bars are Standard Deviations, where *n* = 3.

**Figure 12 ijms-26-01734-f012:**
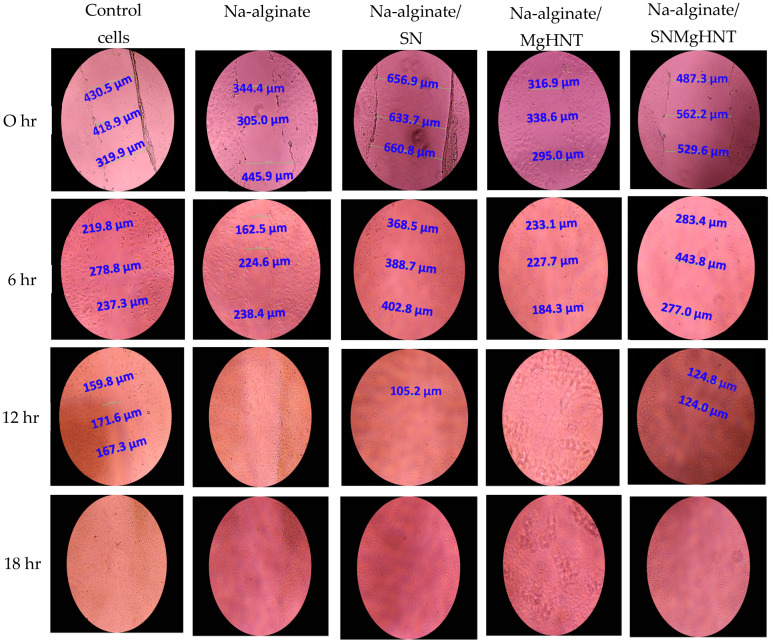
Images of Mechanical Wound Assay (Scratch Assay) with Mouse Embryonic Fibroblast Cell of Nanocomposite Hydrogels for 0 h, 6 h, 12 h, and 18 h. Control Cells are without the Nanocomposites. The numbers in blue are the distance of the wound created in which cells are required to cover. Images were taken with a magnification of 10× set at PH1 condenser and 35 mm focal length.

**Figure 13 ijms-26-01734-f013:**
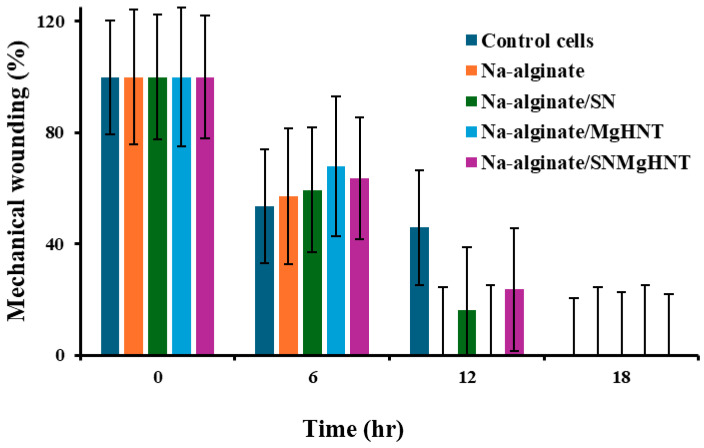
Scratch Assay Quantification of Nanocomposite Hydrogels. Error Bars are Standard Deviations where *n* = 3.

**Figure 14 ijms-26-01734-f014:**
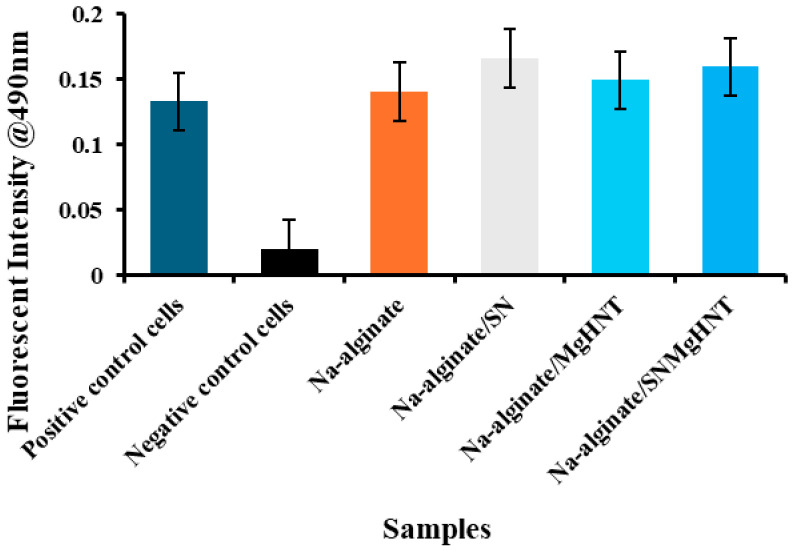
Graphical Representation of Cell Migration Assay of Nanocomposite Hydrogels with Mouse Embryonic Fibroblast Cells. Error Bars are Standard Deviations, where *n* = 3.

**Figure 15 ijms-26-01734-f015:**
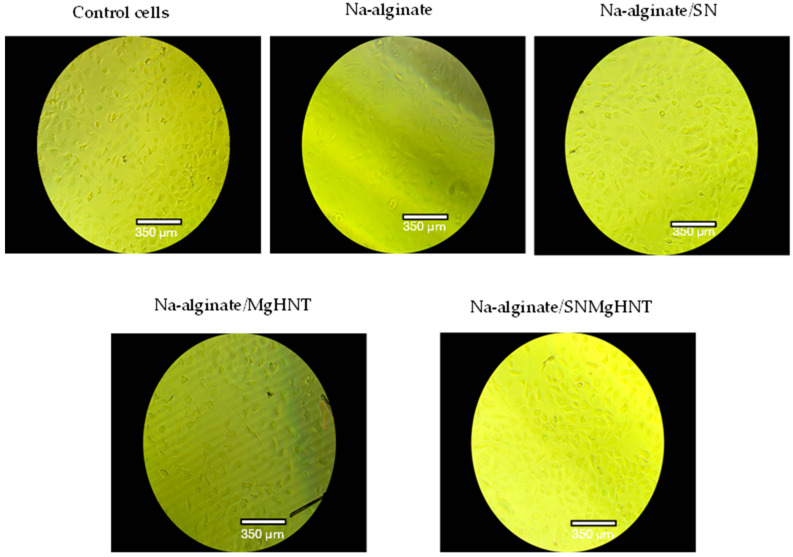
Images of β-Galactosidase Test with Mouse Embryonic Fibroblast Cell of Nanocomposite Hydrogels. Control Cells are without the Nanocomposites.

**Figure 16 ijms-26-01734-f016:**
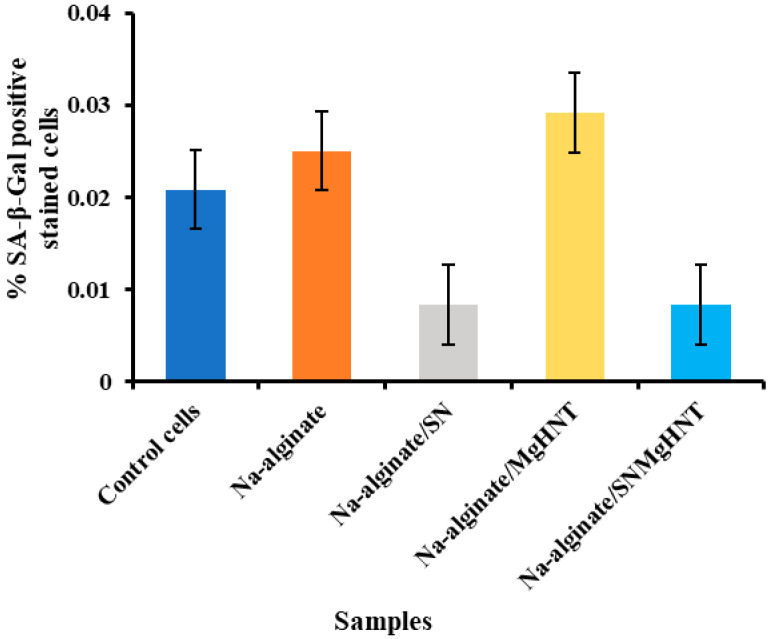
Graphical Representation of % SA-β-Gal Positive Stained Cells of Nanocomposite Hydrogels with Mouse Embryonic Fibroblast Cells. Error Bars are Standard Deviations where *n* = 3.

**Figure 17 ijms-26-01734-f017:**
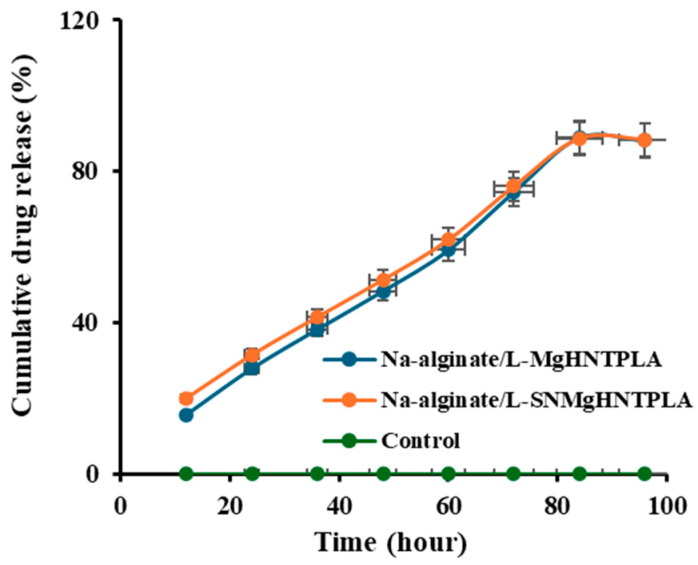
In Vitro Cumulative Release of Gentamicin from Nanocomposite Hydrogels to PBS Buffer Solution at 37 °C.

**Figure 18 ijms-26-01734-f018:**
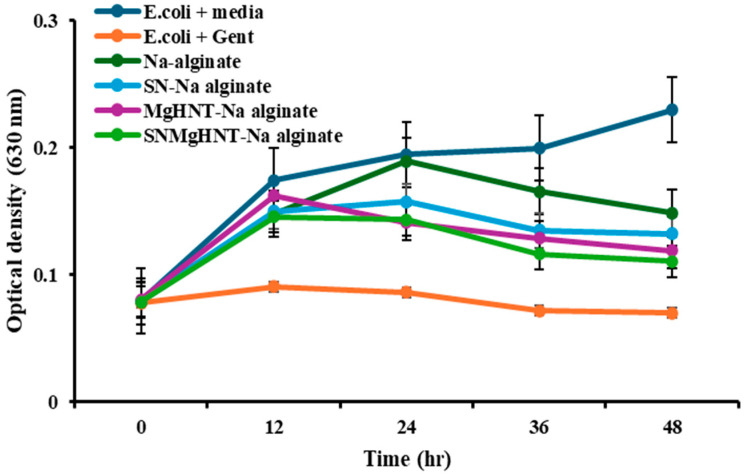
Bacteria Inhibition after 48 h Against *E. coli*. Optical Density was taken at 630 nm wavelength. Error Bars are Standard Deviations, where *n* = 3.

**Figure 19 ijms-26-01734-f019:**
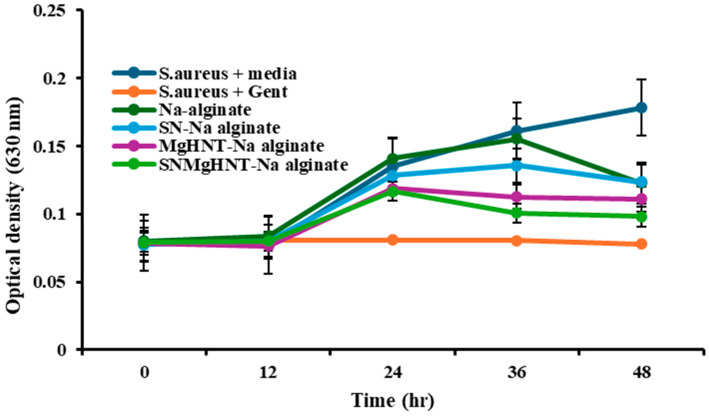
Bacteria Inhibition after 48 h against *S. aureus*. Optical Density was taken at 630 nm wavelength. Error Bars are Standard Deviations, where *n* = 3.

**Figure 20 ijms-26-01734-f020:**
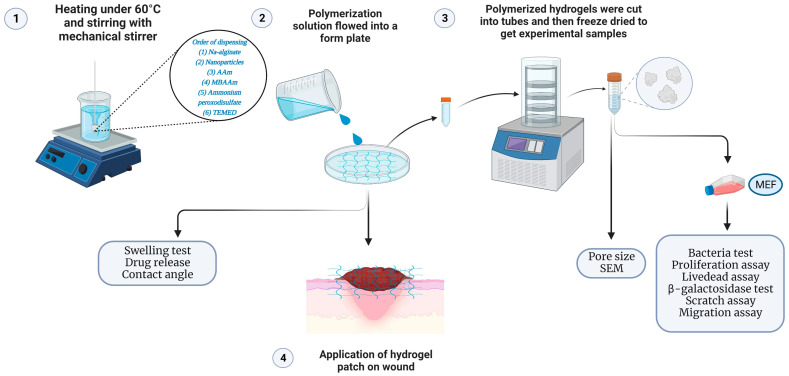
Schematic Illustrating Hydrogels Fabrication and Required Tests. (Figure Created through BioRender.com).

**Figure 21 ijms-26-01734-f021:**
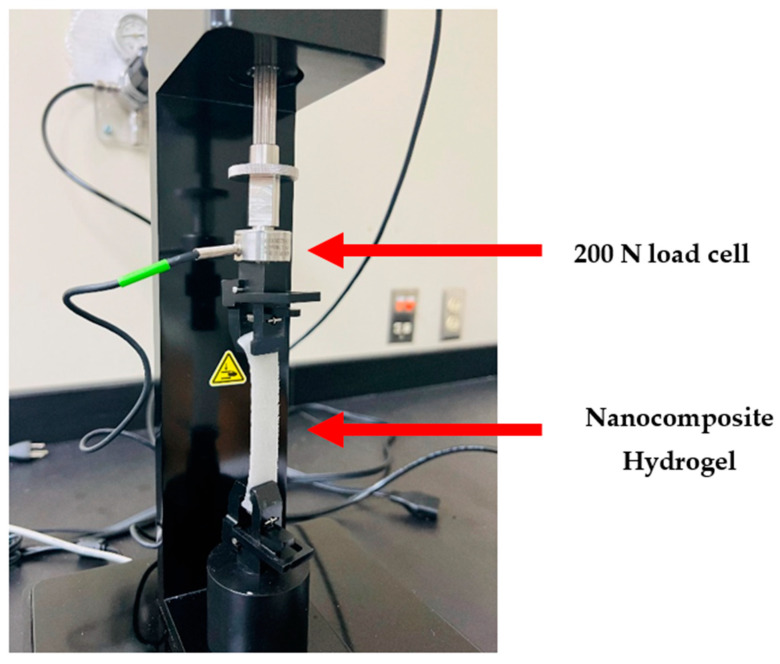
Experimental Setup for Tensile Test.

## Data Availability

Data are contained within the article.

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
