# Peer review of "Fabrication and Characterization of a Stretchable Sodium Alginate Hydrogel Patch Combined with Silicon Nitride and Metalized Halloysite Nanotubes to Develop a Chronic Wound Healing Treatment"

_ijms, 2025, doi:10.3390/ijms26041734_

Round 1
Reviewer 1 Report
Comments and Suggestions for Authors
In this research, the authors present a Stretchable Sodium Alginate Hydrogel Patch combined with Silicon Nitride and Metalized Halloysite Nanotubes which is potential for chronic wound treatment. Overall, the topic is relevant to the scientific community and matches the journal's purpose. However, the current manuscript has plenty of issues at this stage.
1. The paper title can be improved.
2. For the introduction, a paragraph may be added to briefly show the design of the current study.
3. For all the figures, text should be clean. Formats should be consistent. Scale bars are needed for all photos.
4. For Fig10, the data is very rough. Variations of presented data may be discussed.
5. In vivo experiments may be preferred.
6. The limitations of the current study may be discussed in detail.
Comments on the Quality of English LanguageThe English could be improved to more clearly express the research.
Reviewer 2 Report
Comments and Suggestions for Authors
The article ijms-3437119 refers to currently developed, very useful materials – active dressings, which can significantly support the treatment of chronic wounds. The Authors proposed to use for this purpose a nanocomposite sodium alginate hydrogel modified with silicon nitride and magnesium oxide deposited on the halloysite nanotubes. Despite various methods of examining the properties of the obtained film, no significant improvement in the physicochemical or antimicrobial properties of the developed material was demonstrated. The presentation of the purpose and results of the research also requires significant improvement. In my opinion, the article should undergo a major revision.
The most important shortcomings are listed below:
1. Many of the references cited in the Introduction date back 15 years or more. In the Introduction and throughout the article, no reference was made to the risk of undesirable migration of nanoparticles into the treated organism.
2. The Introduction does not contain information about the aim of the research conducted and the assumptions adopted to achieve it. The presentation of the research objective can only be found at the beginning of section 3. Discussion and at the end of section 5. Conclusions.
3. Overall, the structure of the article is disorganized and fragmented. The description of materials and methods used in the research has been moved to the end of the article (between the discussion of results and conclusions).
4. The numbering of figures was not changed when moving them, so the numbering starts from 3 and figures 1 and 2 appear somewhere later in the text (just like the lost figure 11).
5. Some descriptions of the research methods are incomplete and do not allow for a reliable evaluation of the research results (e.g. SEM imaging, tensile strength properties testing).
6. Why is it necessary to provide such detailed descriptions of reagents, including catalog numbers, in the description of the preparation of the tested materials (subsection 4.3), when all these details are provided in subsection 4.1?
7. The photos in Figures 2 and 3 are very blurry and the magnification in the SEM imaging is not uniform. The photos in Figure 8 are of similarly poor quality, and some of the signatures of the materials being assessed are also missing.
8. In general, half of the results presentations are described very briefly in 1-2 sentences, without any commentary (this applies especially to subsections: 2.1.1; 2.1.2; 2.1.4; 2.1.5.; 2.2.2; 2.2.3; 2.2.5 and 2.2.6). In sum, these are only cursory descriptions of what can be seen in the attached graphs or photos. There is a clear lack of commentary and attempts to explain the presented results.
9. The discussion of results in section 3 consists in part of a justification for the research undertaken (which should be in the introduction to the article). Then there are descriptions of the results of selected studies with their fragmentary discussion. This part of the article clearly requires further development.
10. The Authors should consider whether the proposed modifications provided a desirable improvement in the properties of the tested films, especially in relation to the results of the studies on cell proliferation (subsection 2.2.2), scratch treatment (subsection 2.2.3), cell migration (subsection 2.2.4) or drug release (subsection 2.2.6). The presented results have large uncertainty ranges and there are no clear differences between the variants of the tested materials.
11. A significant part of the Conclusions refers mainly to literature data and not to the results of the conducted research. These references should be included in the introduction, while the conclusions should refer to a greater extent to the results of the presented research.
Round 2
Reviewer 1 Report
Comments and Suggestions for Authors
I have no further comments.
Comments on the Quality of English LanguageThe English could be improved to more clearly express the research.
Reviewer 2 Report
Comments and Suggestions for Authors
The manuscript has been significantly improved by the Authors. Following the review guidelines, missing explanations of the research conditions and results were added. The Authors improved the structure of the article and removed unnecessary repetitions. The introduction has been enriched with an indication of the purpose of the research. The research results are now discussed in much greater detail. The conclusions have been rewritten, and the results of the discussed study have been referred to. In my opinion, the article can now be considered for publication in its revised form.